# Multipotent Mesenchymal Stromal Cells from Porcine Bone Marrow, Implanted under the Kidney Capsule, form an Ectopic Focus Containing Bone, Hematopoietic Stromal Microenvironment, and Muscles

**DOI:** 10.3390/cells12020268

**Published:** 2023-01-10

**Authors:** Nataliya Petinati, Irina Shipounova, Natalia Sats, Alena Dorofeeva, Alexandra Sadovskaya, Nikolay Kapranov, Yulia Tkachuk, Anatoliy Bondarenko, Margarita Muravskaya, Michail Kotsky, Irina Kaplanskaya, Tamara Vasilieva, Nina Drize

**Affiliations:** 1Laboratory for Physiology of Hematopoiesis, National Medical Research Center for Hematology, Ministry of Health of the Russian Federation, 125167 Moscow, Russia; 2Department of Immunology, Faculty of Biology, Federal State Budget Educational Institution of Higher Education M.V. Lomonosov Moscow State University, 119234 Moscow, Russia; 3Bioclinic for Working with Animals, Federal State Budgetary Scientific Institution Izmerov Research Institute of Occupational Health, 105275 Moscow, Russia; 4MNIOI Them. P.A. Herzen—Branch of the Federal State Budgetary Institution “NMITs Radiology” of the Ministry of Health of Russia, Department of Pathomorphology, 125284 Moscow, Russia; 5Department of Cell Biology, Faculty of Biology, Federal State Budget Educational Institution of Higher Education M.V. Lomonosov Moscow State University, 119234 Moscow, Russia

**Keywords:** multipotent mesenchymal stromal cells (MSCs), implantation, ectopic foci, differentiation, proliferation

## Abstract

Multipotent mesenchymal stromal cells (MSCs) are an object of intense investigation due to their therapeutic potential. MSCs have been well studied in vitro, while their fate after implantation in vivo has been poorly analyzed. We studied the properties of MSCs from the bone marrow (BM-MSC) before and after implantation under the renal capsule using a mini pig model. Autologous BM-MSCs were implanted under the kidney capsule. After 2.5 months, ectopic foci containing bones, foci of ectopic hematopoiesis, bone marrow stromal cells and muscle cells formed. Small pieces of the implant were cultivated as a whole. The cells that migrated out from these implants were cultured, cloned, analyzed and were proven to meet the most of criteria for MSCs, therefore, they are designated as MSCs from the implant—IM-MSCs. The IM-MSC population demonstrated high proliferative potential, similar to BM-MSCs. IM-MSC clones did not respond to adipogenic differentiation inductors: 33% of clones did not differentiate, and 67% differentiated toward an osteogenic lineage. The BM-MSCs revealed functional heterogeneity after implantation under the renal capsule. The BM-MSC population consists of mesenchymal precursor cells of various degrees of differentiation, including stem cells. These newly discovered properties of mini pig BM-MSCs reveal new possibilities in terms of their manipulation.

## 1. Introduction

Currently, many clinical protocols use multipotent mesenchymal stromal cells (MSCs) for nearly every clinical application imaginable, including graft-versus-host disease, perianal fistulas, neurodegenerative and cardiac disorders, COVID-19, and cancer. Several companies are in the process of commercialization of MSC-based therapies. However, most of the clinical-stage MSC-based therapies are still unable to prove their efficacy in treatment. The therapeutic functions of MSCs administered to humans are not as robust as those demonstrated in preclinical studies, and in general, the realization of cell-based therapy is impaired by a myriad of steps that introduce heterogeneity [1]. Recent advances in stem cell research have given impetus to the development of stem cell therapies that aim to repair or replace damaged tissues and organs.

The MSC population is heterogeneous [2]. Cellular heterogeneity refers to the genetic and phenotypic differences, which reflect their various fate choices, viability, proliferation, self-renewal probability, and differentiation into different lineages. MSCs’ heterogeneity depends on their origin (biological niche) or the specific donors’ features (age, diseases, or other unknown factors). It is accepted that many culture conditions to which MSCs are subjected, such as O_2_ tension, substrate and extracellular matrix cues, inflammatory stimuli or genetic manipulations, can influence their resulting phenotype [3]. Many studies have shown the therapeutic potential of MSCs or their progeny. However, the differentiation and proliferation heterogeneity of MSC constitutes an important barrier for transferring these capabilities to the clinic.

Moreover, the functional properties of MSCs and their differentiation in the organism depend on the method of transplantation, co-transplantation with different cells and the carriers used. For example, combined transplantation of MSCs and endothelial progenitor cells in comparison with transplantation of MSCs alone significantly promoted angiogenesis, bone regeneration, revascularization and tissue repair in cerebrovascular disease; but not in cardiovascular disease [4,5].

For the sake of preserving damaged tissue integrity and providing physical support and trophic supply for tissue regeneration, MSC transplantation on various scaffolds has come to the front stage in therapy for several diseases along with the constant progress of stem cell engineering. Through integrating into the implanted scaffold, MSCs improve its integration by promoting the healing process [6,7,8].

Large animal experiments are important models in pre-clinical studies. Recently, mini pigs have been used in surgical training and regenerative research. Pigs have great biophysical and biochemical similarities to humans [9]. Comparative studies have demonstrated that mini pigs MSCs from various sources, including bone, bone marrow, skin, or adipose tissue, have similar characteristics [10]. For this reason, the pig is an attractive potential model for the study of certain pathologies [11]. In particular, mini pigs are widely used in medical research as a source of organs for xenotransplantation [12,13], as a model of cartilage repair [14,15], and in the field of embryonic stem cells and induced pluripotent stem cells research [16]. In recent years, many researchers have actively sought for techniques to create lines of porcine MSCs from various tissues. Pig MSCs were isolated from bone marrow [17] and other sources [10].

Upon MSC transplantation, the researcher should be able, first, to provide exact cell localization and, second, to monitor their further effectiveness. There are few works devoted to this topic. The well-studied mouse model of the implantation of bone marrow cells under the kidney capsule lacks these problems [18]. However, obtaining mouse MSCs is complicated [19].

Despite detailed descriptions of the properties of porcine MSCs, some analysis have not been done yet. In this work, mini pigs were used to analyze the properties of MSCs upon implantation into the body. The aim of this study was to study the porcine MSC properties in a primary culture derived from bone marrow cells, and in secondary culture, which was established from the focus formed after autologous MSC implantation under the kidney capsule.

## 2. Materials and Methods

### 2.1. Animals

All the animal experimental procedures performed were based on approved guidelines of Regulations for Animal Experiments and Related Activities at Federal State Budgetary Scientific Institution Izmerov Research Institute of Occupational Health. Four healthy mini pig hybrids (Vietnamese bellied and Wiesenau breeds) were 7.5–8.5 months-old at the beginning of the experiments (mean weight 29.7 ± 1.6 kg). These animals were subjected to a nephrectomy at the age of 10–10.5 months (mean weight 42.1 ± 2.8 kg). All manipulations with each mini pig were performed following general anesthesia.

### 2.2. MSC Isolation and Culture

Bone marrow cells were aspirated and mixed with heparin (50 units/mL). For isolation of mononuclear cells, bone marrow (5–7 mL) was mixed with an equal volume of alfa-MEM (Hyclone, Logan, UT, USA) medium containing 0.2% methylcellulose (1500 Cp), Sigma-Aldrich, St. Louis, MO, USA). After 40 min of incubation at room temperature, most erythrocytes and granulocytes had precipitated, while the mononuclear cells remained in suspension. The supernatant fraction was aspirated and centrifuged for 10 min at 450× *g*. The pellet was suspended in a culture medium containing alfa-MEM supplemented with 10% fetal bovine serum (Hyclone, Logan, UT, USA), 2 mM L-glutamine (Hyclone, USA), 100 U/mL penicillin (Ferein, Moscow, Russia) and 50 μg/mL streptomycin (Ferein, Moscow, Russia). The cells were plated at 3 × 10^6^ and 27 × 10^6^ cells per T25 cm^2^ and T175 cm^2^ culture flask (Corning-Costar, NY, USA) respectively. When a confluent monolayer of cells had formed, the cells were washed with sodium chloride physiological solution (Sigma-Aldrich, St. Louis, MO, USA) containing 0.02% EDTA (Sigma-Aldrich, St. Louis, MO, USA), and detached by 0.25% trypsin–EDTA (Sigma-Aldrich, St. Louis, MO, USA). The cells were then seeded at the density of 4 × 10^3^ cells/cm^2^ and maintained at 37 °C and 5% CO_2_. MSCs had been proliferating to the maximum during the first 10 passages, and then the proliferation index had decreased. These cells can be effectively grown for 16 passages.

To obtain individual MSC clones 240 cells were plated 1 cell per well in a 96-well plate, in a culture medium with 20% fetal bovine serum and 5 ng/mL bFGF. After 2 weeks, the number of empty wells in the plate was determined. The cloning efficiency of MSCs was determined by the Poisson formula. For the further analysis, cells from confluent wells were transferred into T25 culture flasks.

### 2.3. Flow Cytometry Activated Cell Sorting (FACS) Analysis of MSC

Phenotyping of MSCs was performed by flow cytometry after 2–5 culture passages. Cells were incubated with monoclonal mouse anti-human antibodies against CD105 (BD Pharmingen, San Jose, CA, USA) and CD90FSP (Fibroblast Surface Protein) (Sigma St. Louis, MO, USA) for 30 min at 40 °C. These primary anti-human antibodies had proven anti-pig activity according to the manufacturer’s datasheets. Then the cells were washed 2 times with phosphate buffer, and then cells were stained with a second goat-anti-mouse antibody conjugated with fluorescein isothiocyanate IgG/IgM (BD Pharmingen, San Jose, CA, USA) for 30 min at 40 °C. Then cells were washed twice with phosphate buffer. The level of nonspecific binding of antibodies was taken into account using an isotype control; for this, cells were incubated with mouse IgG1 immunoglobulins (BD Pharmingen, San Jose, CA, USA). To assess the viability of the analyzed cells, 7-aminoactinomycin D (7-AAD, Sigma, St. Louis, MO, USA) was used. The analysis of fluorescence intensity was carried out on a FACSCalibur device (BD Biosciences, USA), the results were processed using WinMDI v.2.8 (© Joseph Trotter, West Lafayette, IN, USA) and FlowJo v.7.2.4 (Tree Star Inc., Ashland, OR, USA).

### 2.4. Osteogenic and Adipogenic Differentiation

To analyze adipogenic differentiation abilities, 2000–5000 MSCs at passages 2–3 were plated per well in a 6-well plate, in which sterile cover glasses had been previously placed. The culture medium was supplemented with 1 μM dexamethasone (Sigma, St. Louis, MO, USA), 60 μM indomethacine (Sigma, St. Louis, MO, USA) and 5 μg/mL insulin (Sigma, St. Louis, MO, USA).

To analyze osteogenic abilities, 500–1000 MSCs at passages 2–3 were plated per well of a 24-well plate. The culture medium was supplemented with 0.1 μM dexamethasone, 0.15 mM ascorbic acid 2-phosphate trisodium salt (Sigma, St. Louis, MO, USA) and 10 mM glycerol-2-phosphate (Sigma, St. Louis, MO, USA).

The medium was changed twice a week for two weeks. Then, the cells on the glasses were fixed and stained with Oil Red O to detect adipogenic differentiation. Cells in the 24-well plate were carefully washed and stained in the wells with Alizarin Red to detect calcium deposits, which are markers of osteogenic differentiation.

Using the presented methods of differentiation, differentiated cells were obtained from all MSC samples derived from the bone marrow of the mini pigs.

### 2.5. Determination of the Number of MSC Doublings

During continuous passaging, the number of population doublings was calculated using the formula: X = ln(Nf/Ni)ln2, where Ni and Nf are the initial and final numbers of cells, respectively, and ln is the natural logarithm. To yield the cumulated doubling level, the population doubling for each passage was calculated and added to the population doubling levels of the previous passages.

### 2.6. MSC Labeling with a Lentiviral Vector Containing the Green Fluorescent Protein (eGFP) Gene

MSCs were labeled using third-generation LeGo lentivectors containing the eGFP marker gene with a deleted promoter as described [20]. This design of the vector was done in order to avoid the expression of GFP itself, as this protein may be immunogenic and cells expressing GFP can be rejected by the recipient’s immune system [21,22]. Viral stocks were obtained by calcium phosphate transfection of phCMVC-VSV-G (R861), pGpur (R1246), pMDLg/pRRE, and pRSV Rev plasmids into HEK 293T/17 cells. To determine the virus titer, 10,000 cells of 293T/17 cell line and/or porcine MSC (mini pig # 106) were seeded in a 24-well plate, and dilutions of the initial supernatant were made in increments of 10. The virus was added with polybrene 8 pg/mL. Twenty-four hours later, the medium with the virus was changed to a complete culture medium. After another 48 h, DNA was isolated from cells and the virus titer was assessed by real-time PCR using a calibration curve. The calibration curve was plotted based on dilutions of the initial plasmid. The presence of cells carrying the marker gene was counted using PCR analysis. The titer of the virus used for MSC infection was 2–3 × 10^7^.

To infect MSCs with a lentiviral vector, the culture medium was completely removed from the flasks and viral particles were applied to the sublayers at a rate of approximately 3 × 10^7^ per flask with a bottom area of T175 cm^2^ in 7 mL of alpha-MEM medium with 10% fetal calf serum and 8 μg/mL of polybrene (Sigma, St. Louis, MO, USA). After 6 h, the medium was replaced with 25 mL of complete nutrient medium. The efficiency of infection with the virus was determined in individual cultures. Cells were removed from the substrate 2 days after infection, and their number was counted and analyzed using real-time PCR. Based on the data obtained, a calibration curve was built, which was used to determine the proportion of cells carrying the GFP gene. In this way, approximately 60–70% of cultured MSCs were infected. MSCs after infection with lentivector had effectively grown for 12–13 passages.

### 2.7. Gene Expression Analysis

The determination of gene expression level in MSCs was carried out by reverse transcription followed with real-time quantitative polymerase chain reaction (Taq-Man modification) using the Abiprism 7500 device (Applied Biosystems, Waltham, MA, USA).The total RNA was extracted from MSCs at passage 2 using the standard method [23]. cDNA was synthesized using a mixture of random hexamers and oligo(dT) primers. The gene-specific primers and probes were designed by the authors and synthesized by Syntol R&D (Moscow, Russia). The relative gene expression levels (REL) were determined by normalizing the expression of each target gene to the levels of β-actin and GAPDH and calculated using the ΔΔCt method [24] for each MSC sample.

### 2.8. Analysis of the Concentration of Colony Forming Unit Fibroblasts (CFU-F) in the Bone Marrow of Mini Pigs

For the colony-forming unit fibroblast (CFU-F) analysis, 10^6^ and 5 × 10^5^ mononuclear bone marrow cells were cultivated in alpha-MEM supplemented with 20% fetal bovine serum, 2 mM L-glutamine, 100 U/mL penicillin and 50 μg/mL streptomycin. CFU-F concentration was analyzed on day 14 after staining with 4% crystal violet in 20% methanol.

### 2.9. MSC Implantation under Renal Capsule

MSCs were removed with a scraper from the bottom of the flask, were GFP-marked and then unmarked cells were combined, sedimented for 1 min at 1200 rpm, and placed into a syringe with an infusion system in 6% polyglucin (public corporation Biochimik, Saransk, Russia). We used a scraper, rather than removing cells from the substrate using trypsin, in order not to turn MSCs into a single-cell suspension, since it is known that if intercellular contacts are lost, stromal cells do not form a microenvironment after implantation under the renal capsule [25].

Surgical manipulations consisted in opening access to the kidney without pulling it out, making a hole in the capsule and injecting the cells through the infusion system.

### 2.10. Histological Analysis

The pieces of tissues from the implant were decalcified for 3–4 weeks. The samples were then dehydrated through a graded series of ethanol, cleared with xylene and embedded in paraffin. Serial sections (5 µm) were cut. The sections were deparaffinized and stained with hematoxylin–eosin and the Mallory stain for collagen detection. Stained sections were analyzed under a light microscope (Leica, DM6000B, Heerbrugg, Switzerland).

### 2.11. Statistics

Data in tables are presented as mean with standard error. When normal distribution was confirmed, samples were compared using the unpaired Student’s *t* test; otherwise, Mann–Whitney tests were used for unpaired samples, with differences at *p* < 0.05 considered significant.

Bar chart, scatter plots and a Tuckey-style boxplot were built using GraphPad Prism 8.

## 3. Results

### 3.1. BM-MSCs Characteristics

Bone marrow was obtained from four mini pigs, the concentration of CFU-F was analyzed and BM-MSCs were isolated (Table 1). MSCs’ growth parameters, immunophenotype, and number of implanted GFP-marked and unmarked cells are presented in Table 1.

BM-MSCs were cloned one cell per well of 96-well plate, giving a total of 240 cells for each MSC sample. The average cloning efficiency was 0.20 ± 0.05. That is, every fifth cell gave a clone.

The cumulative cell production of BM-MSCs, the growth kinetics (Figure 1A) and the ability to perform adipogenic and osteogenic differentiation were been analyzed (Figure 1B).

The cells revealed linear growth kinetics for up to 15 passages. BM-MSCs were able to differentiate into adipogeneic and osteogeneic lineages after appropriate induction, Figure 1C,D.

By definition, human MSCs must express certain cell-specific markers, including CD105, CD73, and CD90 [26]. Mini pig MSCs do not express CD73 [27]. BM-MSCs highly express CD90, while CD105 expression is weaker (Table 1), which is consistent with the data of other researchers [10]. Therefore, the obtained mini pig BM-MSCs fulfill the general MSC criteria.

### 3.2. BM-MSCs Implantation under the Kidney Capsule

After BM-MSCs were characterized, part of them were labeled with the DNA sequence of the GFP gene. The labeling efficiency was approximately 60–70%. To obtain BM-MSCs for implantation, some of them were removed with trypsin and the number of cells per flask was counted, and some were removed with a scraper without turning the mixture into a single-cell suspension. It is known that the implantation of suspended bone marrow under the murine renal capsule does not result in microenvironment transfer [18]. Stromal cells require intercellular contacts; therefore, MSCs removed with a scraper were used in the work. To estimate the number of implanted BM-MSCs, the average number of trypsinized cells from one flask was multiplied to the total number of flasks. The calculated number of implanted MSCs is presented in Table 1. On average, each mini pig was implanted with (265 ± 53) × 10^6^ unmarked MSCs in combination with (115 ± 26) × 10^6^ GFP-marked ones. All cells were carefully placed in a syringe and injected under the renal capsule through a venous catheter. After 2.5 months, the mini pigs underwent a nephrectomy under anesthesia, leaving the animal alive. Under the kidney capsule, masses of an indefinite shape containing various types of cells were found (Figure 2A). The foci that had formed were analyzed histologically. In each case the implant was cut into pieces; some of them were fixed in 10% neutral formalin for histological analysis. Figure 2B represents a control section of the kidney with an intact connective tissue capsule and shows the marginal area of the implant with connective tissue and blood vessels. In the implant, ossicles are visible, as well as ectopic hematopoietic foci containing stromal cells and an area containing muscle cells (Figure 2C–F). The hematopoietic ectopic foci formation suggest the differentiation of implanted BM-MSCs into all stromal cell types necessary for maintaining hematopoiesis.

Some small pieces of implant approximately 0.3 × 0.3 cm were placed in a six-well plate in alpha-MEM medium with 10% FBS for further cultivation (Figure 3A).

After 5 days of cultivation, mesenchymal stromal cells of various morphologies emerged (Figure 3B). From each well, cells were removed using trypsin, their immunophenotype was determined, and DNA was isolated for GFP gene detection.

Some of the cells that emerged from all the pieces contained the GFP gene (Figure 3C). Thus, these cells originated from labeled, implanted BM-MSCs. The immunophenotype of this cell population was similar to BM-MSCs (data not shown). So, these cells could be considered MSCs and will be further indicated as IM-MSCs.

IM-MSCs were cloned one cell per well of a 96-well plate. The average cloning efficiency of IM-MSCs was doubled in comparison with BM-MSCs (0.40 ± 0.05 versus 0.2 ± 0.05). A total of 225 individual IM-MSC clones were studied.

Confluent individual clones were transferred to T25 flasks. Most of them varied in their morphology (Figure 3D). All clones were analyzed for the presence of the genetic marker. It was shown that 17 out of 94 studied clones carried the GFP marker (18%).

In the study of the differentiation ability of the resulting clones, it turned out that no complete adipogenic differentiation was detected in any of the clones. Under the action of inducers, 67% (63 from 94 studied) of the clones differentiated in the osteogenic direction. The remaining 33% of the clones did not differentiate in either the adipogenic or osteogenic directions. The presence of the marker gene did not affect the differentiation abilities of clones. Therefore, the population of IM-MSC is heterogeneous both morphologically and functionally.

### 3.3. Comparison of BM-MSC and IM-MSC Clones

To assess the presence and proportion of stem and early progenitor stromal cells in implants the clonal efficiency, the proliferative potential and population doubling time of BM-MSC clones were compared with those of IM-MSC clones (Figure 4A–C).

It turned out that clonal efficiency was significantly higher in IM-MSC than in BM-MSC populations (Figure 4A), but there were no significant differences in the cell cycle duration and proliferative potential of these MSC populations (Figure 4B,C). However, IM-MSC clones derived from different animals varied significantly in proliferative characteristics, indicating individual peculiarities of IM-MSCs (Figure 4D,E). For example, in IM-MSCs from mini pig 107, the time of cell cycle was shorter than in all the others. These clones divided faster, while the number of divisions that the cells were able to do was less than in cells from clones of other implants.

The increase in cloning efficiency and in proliferative potential combined with a decreased ability to differentiate led to the assumption that at least some of IM-MSCs acquired features of earlier (maybe close to embryonic) precursors. The expression of three Yamanaka factors is typical for embryonic stem cells while the NES gene is differentially expressed in various stem cells. Real-time PCR was used to analyze the expression of these genes in BM-MSCs and IM-MSCs. In IM-MSCs the expression of the C-MYC and NES genes increased significantly, while the level of KLF4 expression increased slightly and insignificantly. The expression of OCT4 in IM-MSCs was significantly reduced compared to in BM-MSCs of the same pigs (Figure 4F).

## 4. Discussion

The model for the formation of an ectopic focus from implanted cells and tissues makes it possible to study their properties in vivo [28,29]. Various properties of MSCs, such as their radiosensitivity [30,31], aging processes [32], proliferative potential [33,34] and many others, were studied using the method of forming a focus of ectopic hematopoiesis under the renal capsule of mice. Stromal progenitor cells in culture have been effectively studied by implanting an adherent cell layer of long-term bone marrow culture under the mouse kidney capsule [35]. It would be attractive to study the properties of MSCs during their implantation, rather than intravenous administration, since injected intravenously stromal cells practically cannot overcome the vascular endothelium barrier due to the absent or weak expression of CXCR4 [36,37,38]. In most clinical trials, MSCs are transplanted as a cell suspension either intravenously or in combination with various carriers [39,40,41]. There are several works on the treatment of perianal fistulas [42], in which MSCs were implanted without carriers and inductors. However, MSCs are adherent cells and their function depends on intercellular contacts. The loss of these contacts in single-cell suspension may cause the physiological alterations. Therefore, it seems essential to preserve intercellular contacts during MSC transplantation. That is why we implanted BM-MSCs as the combination of a single-cell suspension with a detached adherent layer.

Obviously, the method of ectopic foci formation could be applied to human MSCs only in xenogeneic system. Mini pigs provide the adequate model of studying BM-MSCs during implantation in vivo. Porcine BM-MSCs grow well in culture, have a high proliferative potential, and seem similar to human BM-MSCs. Mini pigs are not linear animals, so only autologous implantation is possible. BM-MSCs can be effectively labeled with a lentiviral vector, and thus it is possible to track their fate in the organism. Implanted under the renal capsule, BM-MSCs are capable of forming a focus consisting of terminally differentiated cells of mesenchymal origin, which include muscle, bone and a functional hematopoietic bone marrow microenvironment. It turned out that newly formed ectopic tissues contain cells with MSC characteristics. Such MSCs migrated out of non-dissociated implant tissue pieces. In other words, the implant contains progenitor cells that can withstand three transfers: from bone marrow to culture, from culture to the organism and from the implant again to culture. Therefore, we can conclude that porcine BM-MSC heterogeneous population reveals high self-renewal ability and probably contains true mesenchymal stem cells.

IM-MSCs do not differ in proliferative potential and population doubling time from BM-MSCs. At the same time, the clonal efficiency of IM-MSCs is significantly higher than that of BM-MSCs (about two times higher, Figure 4A). This suggests that there are twice as many clonogenic cells in the IM-MSC population and, most likely, this population contains earlier precursors. It is known that in culture 75% of human MSCs do not divide, 18% have a low proliferative potential, and only 7% of cells have a sufficiently high proliferative potential [43]. Therefore, the proportion of earlier precursor cells increases in MSCs population after re-transplantation.

The proliferative properties in individual IM-MSCs clones varied between animals. The population doubling time of cloned IM-MSCs in one pig was shorter than in all the others. These clones divided faster, while the number of divisions that the cells were able to do was less than in cells from clones of other implants. The same pattern was earlier observed in human MSCs. Analysis of more than 210 clones derived from MSCs of 18 healthy donors revealed only one clone with a very high proliferative potential that could persist in culture for eight passages [20]. One could suggest that detection of genuine mesenchymal stem cells in culture is an extremely rare event.

The IM-MSC clones’ differentiation potential was greatly altered. The absence of successful adipogenic differentiation and the inability of 33% of the clones to differentiate revealed the heterogeneity of IM-MSCs. There are two possible explanations for the lack of adipogenic differentiation in IM-MSC clones. First, the microenvironment under the kidney capsule could possibly preferentially support the proliferation/retention of osteogenic progenitors rather than adipogenic ones. Second, if the progenitor cell is positioned highly in the hierarchy of mesenchymal progenitor cells, it may not acquire the ability to perform lineage-specific differentiation yet (as it was proved for hematopoietic stem and progenitor cells). Moreover, “stem-like” cells may require additional time to differentiate, more than two weeks, as it is proposed in the in vitro differentiation protocol. The IM-MSC population contains cells with high proliferative potential and probably is not mature enough for efficient adipogenic differentiation. It is known that osteogenic differentiation is characteristic of rather early precursors and deteriorates greatly with the age of the organism, being replaced by adipogenic differentiation [44].

BM-MSCs formed a focus after implantation under the renal capsule and that had caused dramatic changes in IM-MSCs (recapitulation occurred), which was confirmed by diverse analyses. The majority of tissues of mesenchymal origin were found in the implant: bone fragments, muscle tissue and the microenvironment for hematopoiesis. Alterations in expression level of factors that maintain the “embryonic/pluripotent” state were revealed. In IM-MSCs that migrated out of the implant, the expression of the *C-MYC* and *NES* genes significantly increased, while the level of *KLF4* expression increased slightly and insignificantly. The expression of *OCT4* was significantly reduced compared to BM-MSCs from the same mini pig. The *NES* gene was expressed by cells forming vascular and ossification centers during bone formation, while their number increased progressively [45]. The ossicles were found in the implant. In combination with upregulation of the *NES* gene in IM-MSCs it suggests the presence of osteogeneic precursors in this MSC population. An increase in the expression of MYC gene in IM-MSCs and the development of various mesenchymal tissues in the implant allows us to suggest that population of BM-MSCs contain multipotent mesenchymal cells and cells with high proliferative potential.

A third of the clones studied is not able to differentiate in either the osteogenic or adipogenic lineages. We assume that this population includes “young” cells not yet capable of terminal differentiation; therefore, there are no more than 30% of such cells. Gene expression was estimated in the total IM-MSC population; perhaps in connection with this we detected only the most pronounced changes and did not reveal significant changes in the expression of the KLF gene. A decrease in OKT4 expression indicates that IM-MSCs are not true embryonic cells. This assumption is confirmed by the absence of teratomas in implants.

Obviously the molecular mechanisms of the appearance of “young” cells in ectopic foci are not similar to those in induced pluripotent cells [46]. It is known that in the cells of some tumors, the expression of the Yamanaka factors genes is increased [47]. However, IM-MSCs, unlike malignant cells, have a high but limited proliferative potential, which allows us to exclude neoplastic transformation.

It has previously been shown that multipotent adult progenitor cells can be obtained from both postnatal and fetal porcine bone marrow. They have a very high proliferative potential, can perform more than 100 doublings, and do not express CD44, CD45 and MHC I and II classes, while expressing Oct3a mRNA and protein at levels close to those observed in human embryonic stem cells. These cells have telomerase activity, preventing telomere shortening even after 100 doublings. At early passages, these cells differentiate into chondrocytes, adipocytes, osteoblasts, smooth muscle cells, endothelium, hepatocyte-like cells, and neuron-like cells [48]. Adult BM-MSCs did not reveal such extensive embryonic-type features, but the implant formed by these cells probably contains MSCs that have the properties similar to those of embryonic mesenchymal cells.

## 5. Conclusions

Porcine BM-MSCs were implanted into the organism without any carriers or additional inductors. They formed different tissues of mesenchymal origin. It proves their multipotential characteristics in vivo. Moreover, the implanted BM-MSCs revealed proliferation and differentiation heterogeneity. The population of BM-MSC included a subpopulation of cells with a high proliferative potential. Among those, there are cells that have some properties similar to embryonic stem cells. The repeated transfer of mesenchymal stromal precursor cells probably activated a dormant subpopulation of stem cells. We can conclude that the BM-MSC population is heterogeneous and consists of mesenchymal precursor cells of various degree of differentiation including stem cells.

These discovered properties of MSC population require further study, as they reveal new possibilities and limitations for manipulating with these cells.

## Figures and Tables

**Figure 1 cells-12-00268-f001:**
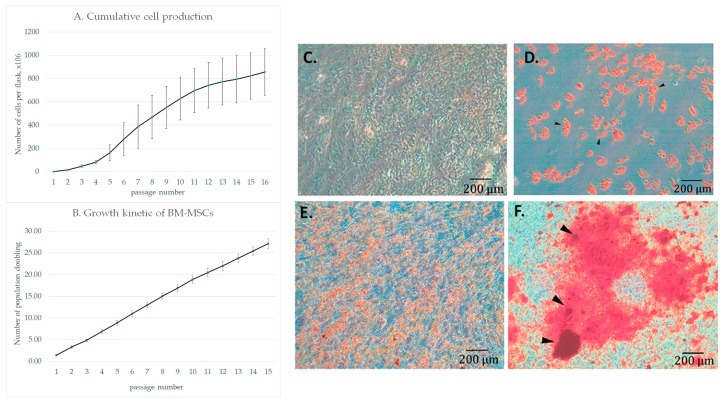
Proliferative and differentiating potential of porcine BM-derived MSCs (BM-MSCs). (**A**) Cumulative cell production. (**B**) MSC Growth Kinetics for 15 Passages. Mean values from four independent samples of BM-MSCs obtained from different animals are presented. (**C**) Control cells stained with Oil Red O. Photo of Pig#106 cells. (**D**) Induction of adipogenic differentiation stained with Oil Red O. Photo of Pig#106 cells. Black arrows point to fat droplets. Magnification 10×. (**E**) Control cells stained with Alizarin Red. Photo of Pig#106 cells. (**F**) Induction of osteogenic differentiation stained with Alizarin red. Photo of Pig#106 cells. Black arrows show calcium deposits. Magnification 10×.

**Figure 2 cells-12-00268-f002:**
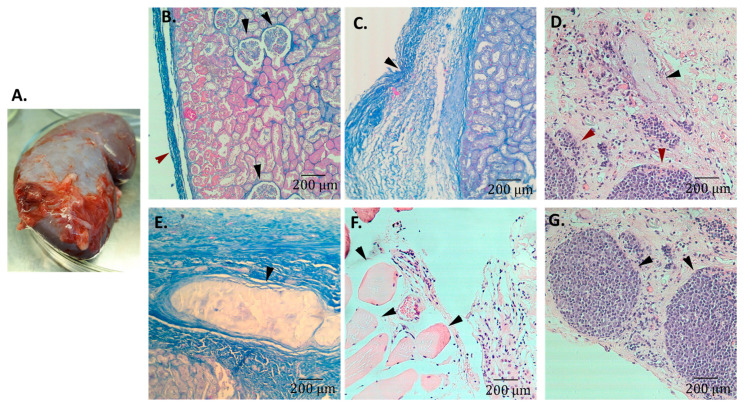
Photos of ectopic foci formed under the kidney capsule of mini pigs after BM-MSCs implantation. (**A**) Overview of the implant. The diameter of the Petri dish is 10 cm. The implant is approximately 2 × 3 cm. Bones are palpable inside. (Pig#103). (**B**) Histological sections of implants. As a control, normal kidney with a connective tissue capsule without an implant is represented. Mallory staining. (Pig#107) Red arrow points to the normal renal capsule. Black arrows indicate renal glomeruli. (**C**) The marginal area of the implant with connective tissue and blood vessels. Mallory staining. (Pig#103) Black arrow points to the renal capsule with part of implant. (**D**) Ossicles (black arrows) and ectopic hematopoietic foci (red arrows) containing stromal cells. Hematoxylin–eosin staining. (Pig#106). (**E**) Ossicle. Mallory staining. (Pig#103). (**F**) Area containing muscle cells. Hematoxylin–eosin staining. (Pig#103). (**G**) Ectopic hematopoietic foci. The area adjacent to the one presented in (**D**) is shown. Hematoxylin–eosin staining. (Pig#106). Magnification 10×.

**Figure 3 cells-12-00268-f003:**
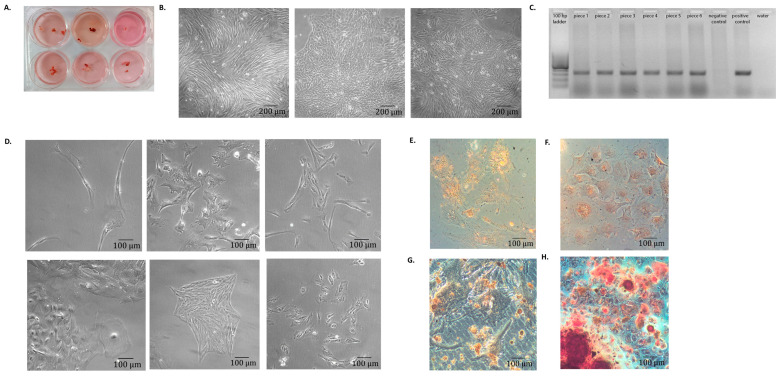
Characteristics of cells that have migrated out of the foci formed under the renal capsule after BM-MSCs implantation. (**A**) Pieces of focus tissue in the wells of a 6-well plate (Pig#99). The approximate size of the pieces was 0.3 × 0.3 cm. (**B**) Morphological heterogeneity of cells that migrated out of the implant (Pig#103). (**C**) Gel electrophoresis of GFP-specific PCR products. DNA of cells that have migrated out of different pieces of the implant was analyzed. (**D**) Cell morphology of various IM-MSC clones (Pig#106). (**E**) Control IM-MSCs stained with Oil Red O. (**F**) IM-MSCs induced to adipogenic differentiation stained with Oil Red O. (**G**) Control IM-MSCs stained with Alizarin red. (**H**) IM-MSCs induced to osteogenic differentiation stained with Alizarin red. Magnification 10×.

**Figure 4 cells-12-00268-f004:**
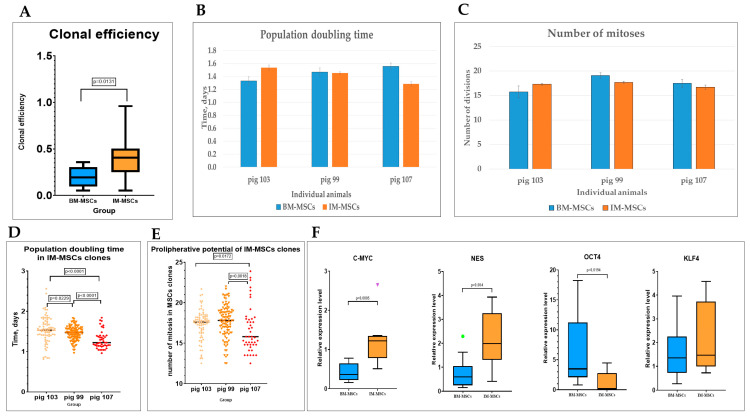
Characteristics of IM-MSCs. (**A**) Comparison of clonal efficiency of BM-MSCs and IM-MSCs. BM-MSCs and IM-MSCs were cloned 1 cell per well of the 96-well plate. Cloning efficiency was determined by the Poisson formula. (**B**) Population doubling time of BM-MSCs and IM-MSCs. (**C**) The number of performed mitoses. Variability of individual clones from different pigs. (**D**) Range of population doubling time in IM-MSCs clones from different mini pigs. (**E**) Individual differences in the proliferative potential of IM-MSCs clones from different mini pigs. (**F**) Comparison of C-MYC, NES, KLF4 and OCT4 relative gene expression in BM-MSCs and IM-MSCs. Data are presented as Turkey style box plots (**A**,**F**), scatter plots (**D**,**E**) and bar graphs (**B**,**C**). Green circle and purple triangle indicate outliers in corresponding groups.

**Table 1 cells-12-00268-t001:** Characteristics of mini pigs, CFU- F and BM-MSCs.

Animal Number	Age (Months)/ Weight (Kg)/ Gender	CFU-F Concentration in the Bone Marrow per 10^6^ Cells	Time to P0, Days	Cumulative MSC Production for 15 Passages, ×10^6^	Immunophenotype MSCs				Number of Implanted under Renal Capsule MSCs, ×10^6^	
					CD90		CD 105			
					MFI	Iso-Type Control	MFI	Isot-Ype Control	MSC	MSC-GFP
106	8/25.2/female	175	7	979,928	133,603	2962	5757	3391	186	38.5
103	7.5/30/male	96.5	11	722,869	122,662	2402	5830	2633	161	67
99	7.5/31.6/female	250	12	1,337,155	124,551	2319	6055	2760	338	169
107	8.5/32.1/female	73	12	386,961	119,444	2882	5020	3420	374	108

## Data Availability

All data associated with this study are available in the main text and through the corresponding author upon request.

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
