# Peer review of "Multipotent Mesenchymal Stromal Cells from Porcine Bone Marrow, Implanted under the Kidney Capsule, form an Ectopic Focus Containing Bone, Hematopoietic Stromal Microenvironment, and Muscles"

_cells, 2023, doi:10.3390/cells12020268_

Round 1

Reviewer 1 Report

In the current manuscript Petinati et al. investigated the effects of implantation of autologous BM-MSCs under the kidney capsule using a mini pig model. After 2.5 months they demonstrated the presence of ectopic foci containing bones, foci of ectopic hematopoiesis, bone marrow stromal cells and muscle cells. The cells migrating out of implant (IM-MSCs) showed a functional heterogeneity as to the proliferation and differentiation potential.

This is a well-written manuscript, in which the claims of authors are supported by the data presented. However, there are some issues, which should be addressed in order to improve the quality of the manuscript:

1.     The authors did not use the Ficoll-gradient for isolation of mononuclear cells from the bone marrow of mini pigs. What was the reason for that and what was the purity of BM-MNCs after the mere sedimentation?

2.     After how many days have the authors performed the first trypsinization i.e. when appeared the first confluent MSCs from MNCs?

3.     Please correct mkM or mkg into µM or µg!

4.     How many MSCs were injected under the renal capsule and in which medium? Were they all eGFP+?

5.     As far as I can see, there were used 4 mini pigs for all experiments, i.e. N=4. Is this number of animals statistically reasonable in order to get reliable data. In other words, did the authors consult a professional statistician?

6.     The authors stated that only 60-70% of MSCs implanted under the kidney capsule were transfected with eGFP. What about the rest of unlabeled cells?

7.     After 2.5 months the authors cloned IM-MSCs and found that 17 out of 94 studied clones carried GFP marker (18%). The question is where did the rest of the cells come from, when not from GFP+ MSCs? Can they be of host origin, because all this happens in an autologous setting? Please, explain!

Author Response

  1. The authors did not use the Ficoll-gradient for isolation of mononuclear cells from the bone marrow of mini pigs. What was the reason for that and what was the purity of BM-MNCs after the mere sedimentation?

We did not use Ficoll-gradient for two reasons:

First, we have been obtaining human MSCs for many years and initially isolated mononuclear cells on LYMPHOPREP, or HISTOPAQUE, or by precipitating erythrocytes in 0.1% methylcellulose. It turned out that MSCs after isolation from the supernatant after incubation with 0.1% methylcellulose grow better and faster. This is probably due to the fact, that within 24-48 hours before the change of medium after the initial seeding, MSCs receive many growth factors from granulocytes remaining in the supernatant. We published several manuscripts with this method (Petinati, N.; Kapranov, N.; Davydova, Y.; Bigildeev, A.; Pshenichnikova, O.; Karpenko, D.; Drize, N.; Kuzmina, L.; Parovichnikova, E.; Savchenko, V. Immunophenotypic characteristics of multipotent mesenchymal stromal cells that affect the efficacy of their use in the prevention of acute graft vs host disease. World J. Stem Cells 2020, 12, 1377–1395, doi:10.4252/WJSC.V12.I11.1377.                 Kuzmina, L.A.; Petinati, N.A.; Parovichnikova, E.N.; Lubimova, L.S.; Gribanova, E.O.; Gaponova, T. V; Shipounova, I.N.; Zhironkina, O.A.; Bigildeev, A.E.; Svinareva, D.A.; et al. Multipotent mesenchymal stromal cells for the prophylaxis of acute graft-versus-host disease: a phase II study. Stem Cells Int. 2012, 1–8, doi:10.1155/2012/968213).

Secondly, this cell isolation method does not require the selection of a specific Ficoll-gradient. For example, a human HISTOPAQUE density: 1.077 g/ml, mouse 1.083 g/ml. Optimal gradient for mini pigs is not described.

By definition, MSCs are plastic adherent cells (Dominici, M.; Le Blanc, K.; Mueller, I.; Slaper-Cortenbach, I.; Marini, F.; Krause, D.; Deans, R.; Keating, A.; Prockop, D.; Horwitz, E. Minimal criteria for defining multipotent mesenchymal stromal cells. The International Society for Cellular Therapy position statement. Cytotherapy 2006, 8, 315–317, doi:10.1080/14653240600855905) and macrophages can remain in the flask until the first passage, however, only cells of non-hematopoietic origin remain by the next passage.

  1. After how many days have the authors performed the first trypsinization i.e. when appeared the first confluent MSCs from MNCs?

Mean time to Р0 was 10.5 days (7, 11, 12, 12). We have added this information to Table 1.

  1. Please correct mkM or mkg into µM or µg!

We made a correction on page 3, lines 114, 145, 146, 148.

  1. How many MSCs were injected under the renal capsule and in which medium? Were they all eGFP+?

Cells were injected in 6% polyglucin (public corporation Biochimik). Description added to methods (Page 5, lines 207-208).

The total number of implanted MSCs and the number of cells labeled with GFP gene injected under the renal capsule are shown in Table 1 (columns 10 and 11). The column 11 indicates the number of cells that have undergone lentiviral infection. Not all of the cells carried the genetic marker. During the infection procedure, lentivirus is not included in all cells, but only in 60-70%.We checked the number of labeled cells in independent cultures against a calibration curve used for determination of the virus titer for each MSC sample, see methods page 4, lines 182-186.

Mini pigs were injected both with cells after infection (GFP labeled cells) and unlabeled cells. Before implanting under the kidney capsule, MSCs were removed from the bottom of the flask with a scraper, since it is known that when the stromal turn into a single cell suspension, the focus under the kidney capsule in mice does not form (Chertkov, J.L.; Drize, N.J.; Gurevitch, O.A. Hemopoietic stromal precursors in long-term culture of bone marrow: II. Significance of initial packing for creating a hemopoietic microenvironment and maintaining stromal precursors in the culture. Exp. Hematol. 1983, 11, 243–248). The number of labeled cells was determined in cultures removed with trypsin and converted into a single cell suspension as a separate experiment.

We suggested that labeling could affect the properties of cells, in particular, the ability to form foci, so some of the injected cells were left unlabeled. For safety reasons in gene therapy, transduced and non-transduced cells are administered simultaneously, for example (B-cell reconstitution after lentiviral vector-mediated gene therapy in patients with Wiskott-Aldrich syndrome. Maria Carmina Castiello et al., Allergy Clin Immunol . 2015 Sep;136(3):692-702.e2. doi: 10.1016/j.jaci.2015.01.035).

Therefore, our experiments were deliberately designed in such a way that the implant could not be formed by marked cells only. The presence of marked cells in the implant proves that the transplanted MSCs were involved in the formation of the foci. We cannot rule out some contribution from locally induced mesenchymal cells. It is known that it is possible to induce local mesenchymal precursors to osteogenesis in mice using bone matrix or extracellular matrix from long-term bone marrow cultures. In this case, small foci are formed (Heterotopic bone formation and induced osteogenesis. Gurevich OA. Gematol Transfuziol. 1989 Aug; 34(8):41-7. PMID: 268473; Induced hematopoietic foci in mice. I. Induction of extraskeletal hematopoietic areas using demineralized tooth matrix .Gurevich OA, SamoÄ­lina NL, MedvinskiÄ­ AL, Gan OI. Gematol Transfuziol. 1990 Feb; 35(2):7-12. PMID: 2332137; Induction of hematopoietic microenvironment by the extracellular matrix from long-term bone marrow cultures. Sadovnikova EY, Deryugina EI, Drize NJ, Chertkov JL. Ann Hematol. 1991 May;62(5):160-4. doi: 10.1007/BF01703141. PMID: 2049461). To exclude the influence of the matrix transferred along with MSCs, we labeled part of the cells with the GFP gene. This procedure was done only to show that the focus was not induced solely from local mesenchyme, but was built from implanted MSCs.

Among the cells migrated from the implant in culture, 18% of GFP-positive clones were obtained (Page 8, line 320), which certainly confirms the direct involvement of implanted MSCs in the construction of the resulting focus.

  1. As far as I can see, there were used 4 mini pigs for all experiments, i.e. N=4. Is this number of animals statistically reasonable in order to get reliable data. In other words, did the authors consult a professional statistician?

When MSCs were implanted under the kidney capsule, all 4 mini pigs implanted with MSCs developed ectopic foci. This, of course, is not very much, but mini pigs are large mammals and, for ethical reasons, we did not conduct additional experiments. Implantation of MSCs under the renal capsule in mini pigs requires complex surgical intervention. At the end of the experiment, it was necessary to perform a nephrectomy. All animals remained alive. All data obtained during the study of MSCs are statistically reliable and approved by the institute's group of statisticians led by Dr. Sergey Kulikov.

  1. The authors stated that only 60-70% of MSCs implanted under the kidney capsule were transfected with eGFP. What about the rest of unlabeled cells?

As we observed marked and unmarked cells in the ectopic foci we suggest that unlabeled cells take main part in the formation of the foci.

  1. After 2.5 months the authors cloned IM-MSCs and found that 17 out of 94 studied clones carried GFP marker (18%). The question is where did the rest of the cells come from, when not from GFP+ MSCs? Can they be of host origin, because all this happens in an autologous setting? Please, explain!

Since induced local MSCs form very small foci in mice and only in the presence of an appropriate inducer (references are given in paragraph 4) it seems most likely that ectopic foci developed from implanted MSCs. We introduced unlabeled cells in relation to labeled cells on average 1 to 3.2, i.e. three times more unlabeled than labeled (Table)

Number of implanted cells

Animal #

MSCs (number of implanted cells, x 106)

MSCs-GFP (number of implanted cells, x 106)

106

186

38.5

103

161

67

99

338

169

107

374

108

Theoretically, we should have received no more than 25% of labeled clones (in fact, less, since when labeling only 60-70% carry GFP), and we got 18% of labeled clones as could be expected (25*0.7 = 17.5). Thus we did not expect that all cells of the ectopic foci would be marked. The fact that the proportion of labeled cells in the foci coincides with the proportion of labeled cells among the implanted MSCs confirms once again that the functional abilities of labeled and unlabeled cells in our experiment are the same. We also received another confirmation that the foci are formed mainly from implanted MSCs, and not induced from the local mesenchyme.

Reviewer 2 Report

The paper of Petinati et al., entitled “MULTIPOTENT MESENCHYMAL STROMAL CELLS FROM PORCINE BONE MARROW, IMPLANTED UNDER THE KIDNEY CAPSULE, FORM AN ECTOPIC FOCUS CONTAINING BONE, HEMATOPOIETIC STROMAL MICROENVIRONMENT, AND MUSCLES” is devoted by the study of the changes in the function of mesenchymal stem cells after passaging them in vivo in an altered microenvironment. The authors found that the MSCs, after transplantation in vivo, acquired higher self-renewal but decreased differentiation abilities and assumed that the cells might have de-differentiated. However, another possible explanation of the observed effects could be clonal selection of self-renewing stem cells. Despite the work is of considerable interest, there are some issues that need to be improved:

11.       The Methods section should be divided into thematic sub-sections.

22.       The reasons for labeling part of MSCs are not clear. In the case of using homogeneously labeled MSCs for transplantation, the Authors can significantly improve their work by demonstrating the origin of the various tissues of the implant (namely, whether they originate from the host or from the donor). Showing the presence of the GFP gene (Figure 3C) by PCR does not imply that all the cells recovered post-transplant were GFP positive and represent the actual transplanted cells. Since it is an autologous transplantation, the recovered cells might have a mix of transplanted and non-transplanted cells.

33. Another question is GFP efficiency was approximately 60-70%. How did they obtain such numbers for primary culture? Did they use several repeats of transduction?

44.       Comparison of BM-MSC and IM-MSC clones can be improved by registration of migratory activity and sensitivity to key hormonal regulators of MSCs.

55.       The Authors widely use the term “functional heterogeneity” with regard to MSCs. It is rather vague concept. What the Authors mean under this term?

66.       The authors suggested that “An increase in the expression of MYC gene in IM-MSCs and development of various mesenchymal tissues in the implant allows to suggest that population of BM-MSCs contain multipotent mesenchymal cells and cells with embryonic-like proliferative potential.” (lines 405-407). But the expression of other embryonic stem cell-related transcription factors OCT4 and KLF4 were either downregulated or did not change in the IM-MSCs versus BM-MSCs. The Authors should discuss this point.

77.       Figure titles should carry information and be the self-explanatory. The normal figure title should give the entire information to the reader even without reading the text. All figure titles should be improved with this in mind.

Author Response

  1. The Methods section should be divided into thematic sub-sections.

We divided the methods section into sub-sections.

  1. The reasons for labeling part of MSCs are not clear. In the case of using homogeneously labeled MSCs for transplantation, the Authors can significantly improve their work by demonstrating the origin of the various tissues of the implant (namely, whether they originate from the host or from the donor). Showing the presence of the GFP gene (Figure 3 C) by PCR does not imply that all the cells recovered post-transplant were GFP positive and represent the actual transplanted cells. Since it is an autologous transplantation, the recovered cells might have a mix of transplanted and non-transplanted cells.

Homogeneous labeled MSCs can only be obtained if the original MSCs are sorted after labeling. In this setting of the experiment, this is impossible.

First, it is known that cells that have been turned into a single-cell suspension and have lost intercellular interactions do not tolerate the hematopoietic microenvironment (Chertkov, J L and Gurevitch, O.A.; Chertkov, J.L.; Gurevitch, O.A. Hematopoietic stem cell and its microenvironment; Medicina: Moscow, 1984;Chertkov, J.L.; Drize, N.J.; Gurevitch, O.A. Hemopoietic stromal precursors in long-term culture of bone marrow: II. Significance of initial packing for creating a hemopoietic microenvironment and maintaining stromal precursors in the culture. Exp. Hematol. 1983, 11, 243–248). Therefore, we needed cells that had not been transformed into a single-cell suspension for implanting under the kidney capsule. We have added explanations to the materials and methods on page 5, lines 208-211.

Second, we used a virus that does not contain a promoter for the expression of the GFP gene, since it is known that the GFP protein can be immunogenic and the cells carrying it are rejected (Stripecke, R.; Del Carmen Villacres, M.; Skelton, D.C.; Satake, N.; Halene, S.; Kohn, D.B. Immune response to green fluorescent protein: implications for gene therapy. Gene Ther. 1999, 6, 1305–1312, doi:10.1038/SJ.GT.3300951. Re, F.; Srinivasan, R.; Igarashi, T.; Marincola, F.; Childs, R. Green fluorescent protein expression in dendritic cells enhances their immunogenicity and elicits specific cytotoxic T-cell responses in humans. Exp. Hematol. 2004, 32, 210–217, doi:10.1016/j.exphem.2003.10.014). So marked cells do not express Green Fluorescent Protein and sorting is impossible. We have added explanations to the text of the article, page 4, lines 165-167 and page 5, lines 208-211. Thus, we initially abandoned the possibility of implanting only labeled cells. We observed the presence of GFP gene in different pieces of implant tissue. It indicated that implanted cells took part in foci formation.

We cannot rule out some contribution from locally induced mesenchymal cells. It is known that it is possible to induce local mesenchymal precursors to osteogenesis in mice using bone matrix or extracellular matrix from long-term bone marrow cultures. In this case, small foci are formed (Heterotopic bone formation and induced osteogenesis. Gurevich OA. Gematol Transfuziol. 1989 Aug; 34(8):41-7. PMID: 268473; Induced hematopoietic foci in mice. I. Induction of extraskeletal hematopoietic areas using demineralized tooth matrix .Gurevich OA, SamoÄ­lina NL, MedvinskiÄ­ AL, Gan OI. Gematol Transfuziol. 1990 Feb; 35(2):7-12. PMID: 2332137; Induction of hematopoietic microenvironment by the extracellular matrix from long-term bone marrow cultures. Sadovnikova EY, Deryugina EI, Drize NJ, Chertkov JL. Ann Hematol. 1991 May;62(5):160-4. doi: 10.1007/BF01703141. PMID: 2049461). To exclude the influence of the matrix transferred along with MSCs, we labeled part of the cells with the GFP gene. This procedure was done only to show that the focus was not induced solely from local mesenchyme, but was built from implanted MSCs.

  1. Another question is GFP efficiency was approximately 60-70%. How did they obtain such numbers for primary culture? Did they use several repeats of transduction?]

We have added a description in the Materials and Methods section on page 4, lines 181-186.

The efficiency of infection with the virus was determined in individual cultures. Cells were removed from the substrate 2 days after infection; their number was counted and analyzed using real-time PCR. Based on the data obtained, a calibration curve was built, which was used to determine the proportion of cells carrying GFP gene.

MSCs were transduced once please see sub-section 2.6 in materials and methods, page 4, lines 177-182.

  1. Comparison of BM-MSC and IM-MSCs clones can be improved by registration of migratory activity and sensitivity to key hormonal regulators of MSCs.

We agree with the remark of the esteemed reviewer. As we know in the absence of inflammation, MSCs practically do not migrate in the body. Previously, we analyzed donor MSCs in the bone marrow of patients who were implanted with MSCs intraosseously and observed donor MSCs approximately at the sites where the cells were injected 6–9 months before analysis (Petinati, N.; Drize, N.; Sats, N.; Risinskaya, N.; Sudarikov, A.; Drokov, M.; Dubniak, D.; Kraizman, A.; Nareyko, M.; Popova, N.; et al. Recovery of donor hematopoiesis after graft failure and second hematopoietic stem cell transplantation with intraosseous administration of mesenchymal stromal cells. Stem Cells Int. 2018, 2018, 6495018, doi:10.1155/2018/6495018). Thus, it did not occur to us to study the ability of MSCs to migrate after implantation and to compare BM-MSCs and IM-MSCs cultures by this property.

With regard to sensitivity to major hormonal regulators, we previously investigated the effect of hydrocortisone on human MSCs and revealed changes in the expression of several genes and clonal efficiency (Shipunova, N.N.; Petinati, N.A.; Drize, N.I. Effect of hydrocortisone on multipotent human mesenchymal stromal cells. Bull. Exp. Biol. Med. 2013, 155, 159–63, doi:10.1007/s10517-013-2102-8). This could be the subject of a separate study, as it may have implications for cell therapy by means of MSCs.

  1. The Authors widely use the term “functional heterogeneity” with regard to MSCs. It is rather vague concept. What the Authors mean under this term?

We meant the diversity of tissues formed by MSCs in the implant, proliferative potential, differentiation stage and ability.

  1. The authors suggested that “An increase in the expression of MYC gene in IM-MSCs and development of various mesenchymal tissues in the implant allows to suggest that population of BM-MSCs contain multipotent mesenchymal cells and cells with embryonic-like proliferative potential.” (lines 405-407). But the expression of other embryonic stem cell-related transcription factors OCT4 and KLF4 were either downregulated or did not change in the IM-MSCs versus BM-MSCs. The Authors should discuss this point.

We added the discussion of this point to the “Discussion”, page 11, lines 435-451.

  1. Figure titles should carry information and be the self-explanatory. The normal figure title should give the entire information to the reader even without reading the text. All figure titles should be improved with this in mind.

We expanded and changed the captions for the figures.

Round 2

Reviewer 2 Report

In the revised version of manuscript of Petinati et al., entitled “MULTIPOTENT MESENCHYMAL STROMAL CELLS FROM PORCINE BONE MARROW, IMPLANTED UNDER THE KIDNEY CAPSULE, FORM AN ECTOPIC FOCUS CONTAINING BONE, HEMATOPOIETIC STROMAL MICROENVIRONMENT, AND MUSCLES” the authors improved some important points. There are still some issues that need to be improved:

1. As far as I understand, GFP labeled cells did not express the GFP protein due to the lack of a promoter. In this case, Authors should not write “part of them were labeled with GFP” (line 257), but should white something like “part of them were labeled with the DNA sequence of the GFP gene”.

2. The term “functional heterogeneity” should be explained in the text of the paper.

Author Response

In the revised version of manuscript of Petinati et al., entitled “MULTIPOTENT MESENCHYMAL STROMAL CELLS FROM PORCINE BONE MARROW, IMPLANTED UNDER THE KIDNEY CAPSULE, FORM AN ECTOPIC FOCUS CONTAINING BONE, HEMATOPOIETIC STROMAL MICROENVIRONMENT, AND MUSCLES” the authors improved some important points. There are still some issues that need to be improved:

  1. As far as I understand, GFP labeled cells did not express the GFP protein due to the lack of a promoter. In this case, Authors should not write “part of them were labeled with GFP” (line 257), but should white something like “part of them were labeled with the DNA sequence of the GFP gene”.

We changed the required phrase (lines 262 and 263).

  1. The term “functional heterogeneity” should be explained in the text of the paper.

We changed the word “functional” to “differentiation and proliferation” (lines 59, 60, 465, 466).